# Artificial neural network analysis of bone quality DXA parameters response to teriparatide in fractured osteoporotic patients

Carmelo Messina[1,2], Luca Petruccio Piodi[3], Enzo Grossi[4], Cristina Eller-Vainicher[5], Maria Luisa Bianchi[6], Sergio Ortolani[6], Marco Di Stefano[7], Luca Rinaudo[8], Luca Maria Sconfienza[1,2], Fabio Massimo Ulivieri[9]*

1 IRCCS Istituto Ortopedico Galeazzi, Milano, Italy, 2 Dipartimento di Scienze Biomediche per la Salute, Università degli Studi di Milano, Milano, Italy, 3 Fondazione IRCCS Ca' Granda Ospedale Maggiore Policlinico, UO Gastroenterologia ed Endoscopia Digestiva, Milano, Italy, 4 Villa Santa Maria Foundation, Centro di Riabilitazioni Neuropsichiatrica, UO Autismo, Tavernerio (CO), Italy, 5 Fondazione IRCCS Ca' Granda Ospedale Maggiore Policlinico, UO Endocrinologia, Milano, Italy, 6 IRCCS Istituto Auxologico, UO Endocrinologia e Malattie del Metabolismo, Milano, Italy, 7 A.O.U. Città della Salute e della Scienza di Torino, Presidio Molinette, Corso Bramante, Torino, Italy, 8 TECHNOLOGIC Srl, Lungo Dora Voghera, Torino, Italy, 9 Fondazione IRCCS Ca' Granda Ospedale Maggiore Policlinico, UO Medicina Nucleare, Milano, Italy

* ulivieri@gmail.com

**Data Availability Statement:** Data are available at Zenodo, DOI: 10.5281/zenodo.3678740, Link: http://zenodo.org/record/3678740#.XlA71BNKiuU.

## Abstract

Teriparatide is a bone-forming therapy for osteoporosis that increases bone quantity and texture, with uncertain action on bone geometry. No data are available regarding its influence on bone strain. To investigate teriparatide action on parameters of bone quantity and quality and on Bone Strain Index (BSI), also derived from DXA lumbar scan, based on the mathematical model finite element method. Forty osteoporotic patients with fractures were studied before and after two years of daily subcutaneous 20 mcg of teriparatide with dual X-ray photon absorptiometry to assess bone mineral density (BMD), hip structural analysis (HSA), trabecular bone score (TBS), BSI. Spine deformity index (SDI) was calculated from spine X-ray. Shapiro-Wilks, Wilcoxon and Student's t test were used for classical statistical analysis. Auto Contractive Map was used for Artificial Neural Network Analysis (ANNs). In the entire population, the ameliorations after therapy regarded BSI (-13.9%), TBS (5.08%), BMD (8.36%). HSA parameters of femoral shaft showed a worsening. Dividing patients into responders (BMD increase >10%) and non-responders, the first presented TBS and BSI ameliorations (11.87% and -25.46%, respectively). Non-responders presented an amelioration of BSI only, but less than in the other subgroup (-6.57%). ANNs maps reflect the mentioned bone quality improvements. Teriparatide appears to ameliorate not only BMD and TBS, but also BSI, suggesting an increase of bone strength that may explain the known reduction in fracture risk, not simply justified by BMD increase. BSI appears to be a sensitive index of TPD effect. ANNs appears to be a valid tool to investigate complex clinical systems.

**Funding:** Author LR was formerly working in Politecnico of Turin and is now employed by the commercial company: "TECHNOLOGIC S.r.l", who provided support in the form of salary for author LR. However, Politenico of Turin and Technologic S.r.l did not have any role in the study design, data collection and analysis, decision to publish, or preparation of the manuscript. The specific roles of this author is articulated in the 'author contributions' section.

**Competing interests:** Author LR was formerly working in Politecnico of Turin and is now employed by the commercial company: "TECHNOLOGIC S.r.l", who provided support in the form of salary for author LR. There are no patents, products in development or marketed products to declare. This does not alter our adherence to PLOS ONE policies on sharing data and materials.

**Abbreviations:** ANNs, artificial neural network analysis; DXA, dual X-ray photon absorptiometry; BMD, bone mineral density; BR, buckling ratio; BSI, bone strain index; CSA, cross sectional area; CSMI, cross sectional moment of inertia; SEC_MOD, section modulus; FS, femoral shaft; HAL, hip axis length; IT, intertrochanter; NN, narrow neck; SD, standard deviation; SDI, spine deformity index; TBS, trabecular bone score; TPD, teriparatide.

## Introduction

Osteoporosis affects more than 75 million people in the United States, Europe and Japan. It causes more than 8.9 million fractures annually worldwide, of which more than 4.5 million occur in America and Europe. The lifetime risk of a wrist, hip or vertebral fracture has been estimated to be in the order of 30% to 40% in developed countries [1].

The diagnosis of osteoporosis is based on the measurement of Bone Mineral Density (BMD) with Dual X-ray Absorptiometry (DXA) [2]. Many studies indicate that the risk of fracture doubles for each standard deviation reduction in BMD [3]. However, assessment of BMD does not completely detect fracture risk. In fact, while the BMD at the spine and at the hip is directly related to the risk of fracture [4], there is an overlap of BMD in patients with or without fractures [5]. This poses a problem for the clinical assessment of fracture risk with BMD alone for its lack of sensitivity [6]. The use of risk factors improves the sensitivity of the assessment [7], but there is a need of further factors in addition to BMD that can predict fracture risk, like the evaluation of bone micro-architectural structure [8]. Its direct examination can be done by an invasive procedure like bone biopsy or indirectly by some non-invasive procedures, like high-resolution peripheral quantitative computed tomography or magnetic resonance [9,10]. These procedures are, however, expensive or with high radiation dose and therefore not suitable for screening. So, there is a need of a simple method for bone micro-architecture and texture analysis. The Trabecular Bone Score (TBS) is a tool correlated with hystomorphometric bone parameters that can be performed during a DXA scan [11]. The TBS evaluates local variations in gray levels from the DXA image of the lumbar spine. It uses experimental variograms of 2D projections images and can discriminate between samples with similar BMD but different 3D trabecular micro-architecture. A high TBS value reflects a good vertebral microarchitectural texture, and *vice versa*. Previous studies showed that the TBS can predict the fracture risk partially independently from BMD [12,13]. Other studies focused on the action of therapies for osteoporosis on TBS. It has been shown that TBS ameliorates less with antiresorptive drugs like bisphosphonates than with bone forming agents like teriparatide [14,15].

Another recently developed bone structural parameter is the lumbar Bone Strain Index (BSI), a stress and deformation vertebral parameter derived from a finite element analysis of the lumbar DXA scan [16,17], that is based on a mathematical model called Finite Element Method (FEM) [18].

BSI calculation is obtained using a constant strain triangular mesh, with the load applied to upper surface and the constraints to the lower. The load applied to the vertebra is specific for each patient and is based on relations between lumbar forces and patient's weight and height provided by Han study [19]. The mechanical properties of the model are defined in a stiffness matrix assigning elastic modulus depending on local BMD according to Morgan's equations [20]. BSI represents the average strain inside the vertebra, obtained with a linear elastic analysis and with the assumption that a higher strain level (high BSI) indicates a greater risk condition. Recent clinical studies found a usefulness of BSI in identifying the osteoporotic patient's subgroup particularly prone to fragility fractures [21] and to characterize young patients affected by secondary osteoporosis [22,23]. BSI increases linearly with stress and vertebral deformation, so its reduction expresses an amelioration of bone status.

Hip Structural Analysis (HSA) is a DXA-derived tool that obtains transverse geometry images acquired by densitometric scans. Its main structural parameters are the area of bone inside the cross sectional area (CSA), the cross-sectional moment of inertia (CSMI) with the section modulus (SECT_MOD) and the buckling ratio (BR), which are correlated to the maximal axial, bending and torsion forces [24]. Hip geometry, as well as hip BMD, has been shown to have an independent correlation with the risk of hip fracture [2,25].

Teriparatide (TPD, 1–34 recombinant human parathyroid hormone) is approved for the treatment of primary and glucocorticoid-induced osteoporosis in patients with high fracture risk [26,27]. A significant increase in TBS and spine BMD was reached with a two years treatment with TPD in post-menopausal women [27,28]. Other studies with TPD demonstrated a significant increase in proximal femur BMD, in a geometric parameter as average cortical thickness, and in the outer- and endo-cortical diameters [29]. There was also an increase of SEC_MOD and a reduction of BR [30–32]. Another work found that TBS significantly increased with TPD, but did not significantly change with alendronate in glucocorticoid induced osteoporosis [33].

Osteoporosis is a multi-factorial pathology, characterized by plenty of variables, which are connected in a complex way that is difficult to investigate with classical standard statistical methods. To approach the complexity of the problem we have employed a new methodology based on an Artificial Neural Network (ANNs). ANNs are computational adaptive systems inspired by the functioning processes of the human brain particularly adapted to solve non-linear problems and to discover subtle trends and associations among variables (32,33). Based on their learning through an adaptive way (i.e., extracting from the available data the information needed to achieve a specific aim and to generalize the acquired knowledge), the ANNs appear to be a powerful tool for data analysis in the presence of relatively small samples.

In this paper we have used a special kind of ANN architecture, the Auto Contractive Map (AutoCM)[34,35]. This method of data mining is a new analytical process able to create a semantic connectivity map in which non-linear associations are preserved, connections schemes are explicated and the complex dynamics of adaptive interactions is captured. The AutoCM approach has been applied in recent years to the analysis of a growing number of different clinical diseases, demonstrating its value in identifying significant associations between clinical, serological and novel "omics" biomarkers [21,36–38]. Therefore, ANNs could be a useful way to better understand the relationships between the numerous different variables that play a role in osteoporosis. So, the aim of this study was to evaluate both with standard statistical and ANNs analysis the DXA bone quantity and quality parameters before and after treatment with TPD in fractured osteoporotic patients.

## Patients and methods

### Patients

In this retrospective study 40 osteoporotic patients (29 women and 11 men) with multiple vertebral osteoporotic fractures and treated with TPD were analyzed. All the women were in post-menopause. The patients were followed at the "IRCCS Fondazione Ca' Granda Ospedale Maggiore Policlinico", Milan, Italy and at the "Istituto Auxologico Italiano IRCCS", Milan, Italy.

All patients underwent a clinical examination, a spine X-ray exam to assess the spine deformity index (SDI) [39,40], and a DXA exam to quantify hip and lumbar BMD, lumbar spine TBS and BSI, and HSA. Patients were treated with daily subcutaneous 20 mcg of TPD (Forsteo, Eli Lilly Company, Indianapolis, IN, USA) for 2 years. At the end of the treatment all patients were assessed again with clinical examination, DXA and spine X-ray.

All the patients signed a written informed consent and local Ethical Committee approval was obtained (Ethics Committee: Milano Area 2. Protocol N 2.0 BQ. 265_2017, 13th June 2017).

### Methods

**DXA data acquisition.** Bone status was investigated with DXA (Hologic Discovery A, Waltham, MA, USA, software version 13.3.0.1), according to the International Society for

Clinical Densitometry (ISCD) guide lines [41]. All patients underwent two scans, a L1-L4 spine scan and a hip scan. Fractured vertebrae were excluded from the analysis. TBS and BSI were automatically obtained from the spine scans, while HSA was automatically obtained from the hip DXA scan in three different regions: Narrow Neck (NN), Intertrochanteric Region (IT) and Femoral Shaft (FS). Finally, Hip Axis Length (HAL, mm) and Shaft Neck Angle (degrees) were also automatically measured from the hip scan. Beside BMD (g/cm2), in every hip region the following parameters were considered: CSA (cm2), CSMI (cm4), BR, width (mm), and the section modulus (SECT_MOD, cm3), which are related to axial and torsion strength. SECT_MOD is derived from Cross Sectional Moment of Inertia (CSMI), that measures the torsional and bending strengths contribution in relationship to the distance from the center to the outer cortical of the considered section. BR is the ratio between the radius (the maximum distance between the center of the bone section and its outer cortical) and the mean thickness of the cortical. It provides a compressive and torsional loads cortical stability index.

**Other data acquisition.** In order to investigate the presence of vertebral fractures all patients were imaged with antero-posterior and lateral X-ray of the spine at the beginning and at the end of the pharmacological treatment. Finally, the SDI before and after therapy was calculated using the semi-quantitative approach [42,43].

**Statistics.** We constructed a semantic connectivity map through Auto-CM system (Semeion), a fourth generation ANNs, to offer some insight regarding the complex biological connections between variables on study. The system highlights the natural links among variables with a graph based on minimum spanning tree theory, where distances among variables reflect the weights of the ANN after a successful training phase. The Auto-Contractive Map (Auto-CM) was born as a new ANNs and was designed at the Semeion Research Center [44,45]. The Auto-CM system finds, by a specific learning algorithm, a square matrix of weighted connections among the variables of any dataset. This matrix of connections presents many suitable features: a) non linear associations among variables are preserved; b) connections schemes among clusters of variables are captured; c) complex similarities among variables become evident. Once an Auto-CM weights matrix is obtained, it is then filtered by a minimum spanning tree (MST) algorithm generating a graph whose biological evidence has already been tested in the medical field [3–6]. The ultimate goal of this data mining model is to discover hidden trends and associations among variables, since this algorithm is able to create a semantic connectivity map in which non linear associations are preserved and explicit connection schemes are described. This approach shows the map of relevant connections between and among variables and the principal hubs of the system. Hubs can be defined as variables with the maximum amount of connections in the map. From a mathematical point of view the specificity of Auto-CM algorithm is to minimize a complex cost function with respect to the traditional ones.

Traditional minimization cost function:

$$E = Min\left\{ \sum_i^N \sum_j^N \sum_q^M u_i^q \cdot u_j^q \cdot \sigma_{i,j} \right\}$$

Auto-CM minimization cost function:

$$E = Min\left\{ \sum_i^N \sum_j^N \sum_k^N \sum_q^M u_i^q \cdot u_j^q \cdot u_k^q \cdot A_{i,j} \cdot A_{i,k} \right\};$$

$$\boldsymbol{A} = \left( 1.0 - \frac{\boldsymbol{w}}{C} \right);$$

$$N = \text{Number of Variables (Columns)};$$

$$M = \text{Number of Patterns (Rows)}.$$

Comparing the two cost functions it is evident how the traditional minimization includes only second order effects, while the Auto-CM considers also a third order effect. Practically, this means that the Auto-CM algorithm is able to discover variable similarities completely embedded in the dataset and invisible to the other classical tools. This approach describes a context which is typical of living systems, where a continuous time dependent complex change in the variable value is present. Auto-CM can also learn under difficult circumstances such as when the connections of the main diagonal of the second connections matrix are removed. When the learning process is organized in this way, Auto-CM identifies specific relationships between each variable and all others. Consequently, from an experimental point of view, it appears that the ranking of its connections matrix is equal to the ranking of the joint probability between each variable and the others. Auto-CM requires a training phase necessary to learn how variables are interconnected. The learning algorithm of CM can be summarised in four orderly steps: a) signal transfer from the input into the hidden layer; b) adaptation of the connections value between the input layer and the hidden layer; c) signal transfer from the hidden layer into the output layer; d) adaptation of the connections value between the hidden layer and the output layer. The MST represents what could called the 'nervous system' of any dataset. In fact, summing up all the connection strengths among all the variables, we get the total energy of that system. The MST selects only the connections that minimize this energy, i.e. the only ones that are really necessary to keep the system coherent. Consequently, all the links included in the MST are fundamental, but, on the contrary, not every 'fundamental' link of the dataset need to be in the MST. Such limit is intrinsic to the nature of MST itself. Every link that gives rise to a cycle into the graph, that destroys the graph's 'treeness', is eliminated, whatever its strength and meaningfulness. To fix this shortcoming and to better capture the intrinsic complexity of a dataset, it is necessary to add more links to the MST, according to two criteria: (1) the new links have to be relevant from a quantitative point of view; (2) the new links have to be able to generate new cyclic regular microstructures, from a qualitative point of view. The additional links superimposed to MST graph generate a Maximally Regular Graph (MRG).

MRG is the graph whose hubness function attains the highest value among all the graphs generated by adding back to the original MST, one by one, the connections previously skipped during the computation of the MST itself. In other words, starting from the MST, the MRG, presenting the highest number of regular microstructures, highlights the most important connections of the dataset. The resulting "diamond" expresses the complexity core of the system and, in our specific case, the core of the syndrome.

AutoCM maps and minimum spanning tree have been applied to the entire population of the study and also in two subgroups, "responders" and "non-responders" to the therapy, established on the basis of a BMD criterium taken from the TPD pivotal registration trial [27], in which the cut-off value of response was a 10% increase in lumbar spine BMD after treatment. On the two groups, the Maximally Regular Graph (MRG) algorithm was then applied to the Spanning Tree. This algorithm introduces new and more complex connections between variables not directly related in the spanning tree [34]. The resulting four maps show the relations of the studied variables in the "responders" before (PreR) and after (PostR) therapy and the same in the "non-responders" (preNR and PostNR, respectively).

Regarding the classic statistical analysis, we first assessed the normality of data using the Sha-piro-Wilks Test, and when the assumption was met (p < 0.05), data were presented as mean with standard deviation. When normality was not satisfied, variables were presented as median with interquartile range (IQR). The comparison of data before and after the treatment was per-formed for all patients as well as for the subgroups (responders and non-responders). Student's t-test was used for data with a normal distribution, while for non-parametric data the Wilcoxon signed rank test was used. A p value lower than 0.05 was considered statistically significant.

## Results

In the entire population the mean age at the enrollment was 70 years ± 10.6 SD (range: 43–91). Patients' BMI before treatment was 25.9 ± 4.09, while after treatment it was 26.1 ± 4.6 kg/m2 (p = 0.542). Table 1 shows mean, median, SD, IQR, variation percentage and statistical significance values of SDI and DXA parameters in the entire population, before and after TPD treatment.

Bone quality parameters BSI and TBS presented an amelioration after treatment, with a vari-ation of -13.9% for BSI and 5.08% for TBS. BMD, which is the bone quantity parameter, showed a significant increase of 8.36%. Hip geometry indexes of femoral shaft, CSA, SECT_MOD and BR, worsened after treatment (-0.98%, -2.33%, 1.62%, respectively), while its BMD ameliorated (0.23%). In our population 14 patients were "responders" and 26 were "non-responders". Inter-estingly, the percentage of gender composition was different between the two groups. In fact, while "non responder" group was mainly composed by women (21/26, about 80%), the "responder" group was quite balanced between males and females (8/14 females, about 55%). When considering only the "responder" population, BMD showed a statistically significant increase of +20.04%, while TBS and BSI showed a variation of +11.87% and -25.46% respec-tively, which were both statistically significant. The only HSA parameter that showed a signifi-cant variation was FS_CSMI (p = 0.01). On the contrary, when considering the "non-responder" population, BSI was the only bone quality parameter showing a statistically signifi-cant variation of -6.75%; neither BMD nor TBS showed a significant change. For HSA, a statisti-cally significant change was found for all the FS parameters (CSMI, SECT_MOD and BR).

Table 2 shows mean/median values and the significance of the variation before and after therapy of SDI and of the DXA parameters in the "responders" to TPD treatment.

Table 3 shows mean/median values and the significance of the variation before and after TPD treatment of SDI and the DXA parameters in the "non-responders". The two groups, "responders" and "non-responders", share the significant modification of only two variables: BSI and FS_CSMI.

Fig 1 shows the connectivity map of all variables linked to the densitometric status before TPD treatment, showing a spread around two nodes, FS_BMD and HAL, that appear to be hubs. The distribution of nodes and their connections after therapy are showed in Fig 2, where NN_SECT_MOD, correlating with bending and torsion resistance, gains a central position. Fig 3A and 3B show the ANNs maps of the "responders" before and after therapy, where the connections between variables increase after TPD, and FS_CSMI becomes the central hub. Fig 4A and 4B are the "non-responders" maps before and after treatment, showing a noticeable paucity of interconnections, particularly before therapy. FS_CSMI looses its hub position in favor of FS_CSA, index related to axial strength.

## Discussion

In this study a population of patients with osteoporosis' fragility fractures treated with subcuta-neous daily injections of TPD, an osteoinductive agent known to ameliorate both mineral den-sity and bone structure, was investigated before and after therapy. In the entire population

**Table 1. SDI and DXA parameters of the entire population (40 patients) before and after TPD therapy.**

|  |  | Mean/median | SD/IQR | Variation % | p-value |
|---|---|---|---|---|---|
| NN_BMD | Before TPD | 0.740 | 0.112 | 2.92% | 0.343 |
|  | After TPD | 0.761 | 0.205 |  |  |
| NN_CSA | Before TPD | 2.347 | 0.401 | 1.62% | 0.312 |
|  | After TPD | 2.385 | 0.454 |  |  |
| NN_CSMI˚ | Before TPD | 2.197 | 1.112 | 1.87% | 0.823 |
|  | After TPD | 2.239 | 1.147 |  |  |
| NN_WIDTH | Before TPD | 3.345 | 0.352 | 0.27% | 0.72 |
|  | After TPD | 3.354 | 0.384 |  |  |
| NN_SECT_MOD | Before TPD | 1.213 | 0.302 | 1.33% | 0.426 |
|  | After TPD | 1.229 | 0.333 |  |  |
| NN_BR˚ | Before TPD | 13.480 | 3.930 | -4.67% | 0.214 |
|  | After TPD | 12.851 | 4.315 |  |  |
| IT_BMD | Before TPD | 0.742 | 0.147 | 1.57% | 0.374 |
|  | After TPD | 0.753 | 0.154 |  |  |
| IT_CSA | Before TPD | 4.125 | 0.943 | 1.65% | 0.365 |
|  | After TPD | 4.193 | 1.008 |  |  |
| IT_CSMI˚ | Before TPD | 11.548 | 6.546 | 0.84% | 0.635 |
|  | After TPD | 11.645 | 6.863 |  |  |
| IT_WIDTH | Before TPD | 5.847 | 0.716 | 0.02% | 0.985 |
|  | After TPD | 5.848 | 0.734 |  |  |
| IT_SECT_MOD˚ | Before TPD | 3.626 | 1.580 | -2.32% | 0.302 |
|  | After TPD | 3.541 | 1.832 |  |  |
| IT_BR | Before TPD | 11.439 | 3.549 | -3.25% | 0.79 |
|  | After TPD | 11.068 | 3.578 |  |  |
| FS_BMD | Before TPD | 1.262 | 0.349 | 0.23% | 0.03* |
|  | After TPD | 1.265 | 0.358 |  |  |
| FS_CSA | Before TPD | 3.624 | 1.239 | -0.98% | 0.034* |
|  | After TPD | 3.588 | 1.282 |  |  |
| FS_CSMI | Before TPD | 3.568 | 1.514 | -1.27% | 0.125 |
|  | After TPD | 3.522 | 1.835 |  |  |
| FS_WIDTH | Before TPD | 3.076 | 0.269 | 0.26% | 0.446 |
|  | After TPD | 3.084 | 0.285 |  |  |
| FS_SECT_MOD | Before TPD | 2.256 | 0.840 | -2.33% | 0.005* |
|  | After TPD | 2.203 | 0.855 |  |  |
| FS_BR˚ | Before TPD | 3.558 | 1.285 | 1.62% | 0.014* |
|  | After TPD | 3.615 | 1.190 |  |  |
| SHAFT_NECK_ANGLE | Before TPD | 128.895 | 5.141 | 0.25% | 0.545 |
|  | After TPD | 129.222 | 5.159 |  |  |
| SDI˚ | Before TPD | 8.000 | 6.000 | 12.50% | 0.068 |
|  | After TPD | 9.000 | 6.000 |  |  |
| HAL | Before TPD | 107.750 | 10.473 | 0.32% | 0.432 |
|  | After TPD | 108.100 | 10.397 |  |  |
| BMD | Before TPD | 0.745 | 0.143 | 8.36% | <0.001* |
|  | After TPD | 0.807 | 0.165 |  |  |
| TBS˚ | Before TPD | 1.123 | 0.159 | 5.08% | 0.019* |
|  | After TPD | 1.180 | 0.158 |  |  |

(*Continued*)

**Table 1.** (Continued)

| | | Mean/median | SD/IQR | Variation % | p-value |
|---|---|---|---|---|---|
| **BSI** | Before TPD | 2.472 | 0.691 | -13.90% | <0.001* |
| | After TPD | 2.129 | 0.666 | | |

˚ = non-parametric distribution, with values presented as median with interquartile range (IQR) and compared with Wilcoxon signed rank test

* = statistically significant difference (p<0.05).

studied both TBS and BSI, indexes of bone texture and strain, respectively, showed a significant amelioration, as well as BMD. For the last it is an expected and well-known result, while there are only few data [46,47] about TBS, confirming its increase after TPD as reported in other previous works [48]. As regards BSI, no data are available about its response to osteoporosis' treatment. In our study the value of BSI decreased significantly after TPD (-13.9%), and this finding is compatible with an increase in bone strength. In fact, the amelioration of bone structure and bone strength related to the vertebra increases the ability of the vertebra to support an external load, and thus reduces the internal strain. Being BSI the representation of the internal strain of the vertebra, a lower value indicates a lower stress and strain level affecting the vertebra, and consequently a lower fracture risk.

Considering the two groups in which the patients were divided, we notice that the only significant difference in "non-responders" was the amelioration of BSI. Of note, differently from "non responders", the "responders" group was balanced in the gender composition with almost the same percentage of both sexes. Thus, gender may have a certain specific impact on BSI variations, despite this result needs to be confirmed in larger samples.

Regarding HSA parameters, significant variations were shown only at the femoral shaft, but they are of very small absolute entity and so of doubtful clinical relevance. Differently to our results, previous studies conducted in a larger set of patients did not found effects of TPD on femoral shaft [30,32], but only on the other femoral regions. Overall, the significant variations of both quantitative (BMD) and qualitative (TBS, BSI, femoral shaft HSA) DXA derived parameters after TPD treatment are consistent with an improvement of bone resistance to mechanical stresses[49,50].

The great number of variables considered in our study can complicate the comprehension of the meaning of the correlations we found, and for this reason we also used an innovative approach to statistical analysis, which is commonly used in artificial intelligence systems, namely the neural network analysis (ANNs) with a potent data mining system. We can define data mining as Data mining extraction of interesting (non-trivial, implicit, previously unknown and potentially useful) patterns or knowledge from a huge amount of data. In medical field data mining represents a relatively new philosophy emerging with the advent of genomic and functional data. The available techniques offered by classical statistics like Principal Component Analysis of Hierarchical clustering suffer from a number of drawbacks due to the complexity of possible interactions between risk factors, their non-linear influence on the disease occurrence and the considerable stochastic components. The more common algorithms of linear projections of variables require generally a Gaussian distribution of data and have limited power when the relationships between variables are non linear. Application of these methods may loose important informations, and establish precise associations among variables having only the contiguity as a known element is difficult. Another possible limitation of currently used statistical methods is that mapping is generally based on a specific kind of "distance" among variables (e.g. Euclidean, City block, correlation, etc), and gives origin to a "static" projection of possible associations. In other words, the intrinsic dynamics due to active

**Table 2. SDI and DXA parameters of the "responders" (14 patients; 8 females and 6 males) before and after TPD therapy.**

| | | Mean/median | SD/IQR | Variation % | p-value |
|---|---|---|---|---|---|
| NN_BMD | Before TPD | 0.706 | 0.103 | 3.56% | 0.411 |
| | After TPD | 0.732 | 0.164 | | |
| NN_CSA | Before TPD | 2.352 | 0.361 | 5.32% | 0.169 |
| | After TPD | 2.477 | 0.518 | | |
| NN_CSMI | Before TPD | 2.534 | 0.724 | 5.97% | 0.062 |
| | After TPD | 2.686 | 0.872 | | |
| NN_WIDTH | Before TPD | 3.513 | 0.376 | 2.28% | 0.093 |
| | After TPD | 3.593 | 0.389 | | |
| NN_SECT_MOD | Before TPD | 1.242 | 0.263 | 5.39% | 0.146 |
| | After TPD | 1.309 | 0.353 | | |
| NN_BR˚ | Before TPD | 14.731 | 2.821 | 2.69% | 0.972 |
| | After TPD | 15.127 | 7.245 | | |
| IT_BMD˚ | Before TPD | 0.691 | 0.130 | 6.02% | 0.196 |
| | After TPD | 0.732 | 0.140 | | |
| IT_CSA | Before TPD | 4.109 | 0.877 | 6.49% | 0.128 |
| | After TPD | 4.376 | 1.017 | | |
| IT_CSMI | Before TPD | 15.269 | 6.191 | -6.00% | 0.184 |
| | After TPD | 14.353 | 8.014 | | |
| IT_WIDTH | Before TPD | 6.251 | 0.791 | 0.21% | 0.822 |
| | After TPD | 6.264 | 0.810 | | |
| IT_SECT_MOD | Before TPD | 3.869 | 1.066 | 3.28% | 0.425 |
| | After TPD | 3.996 | 1.080 | | |
| IT_BR | Before TPD | 12.422 | 2.337 | -4.55% | 0.251 |
| | After TPD | 11.856 | 2.364 | | |
| FS_BMD | Before TPD | 1.237 | 0.245 | 0.29% | 0.895 |
| | After TPD | 1.241 | 0.247 | | |
| FS_CSA | Before TPD | 3.801 | 0.921 | 0.59% | 0.753 |
| | After TPD | 3.823 | 0.927 | | |
| FS_CSMI | Before TPD | 4.239 | 1.480 | 0.33% | 0.01* |
| | After TPD | 4.253 | 1.546 | | |
| FS_WIDTH | Before TPD | 3.210 | 0.281 | 0.28% | 0.506 |
| | After TPD | 3.219 | 0.280 | | |
| FS_SECT_MOD | Before TPD | 2.447 | 0.919 | -0.98% | 0.795 |
| | After TPD | 2.423 | 0.764 | | |
| FS_BR˚ | Before TPD | 3.629 | 1.322 | 2.91% | 0.650 |
| | After TPD | 3.734 | 0.981 | | |
| SHAFT_NECK_ANGLE | Before TPD | 131.021 | 5.185 | -0.23% | 0.709 |
| | After TPD | 130.721 | 4.392 | | |
| SDI | Before TPD | 8.000 | 5.000 | 12.50% | 0.341 |
| | After TPD | 9.000 | 4.500 | | |
| HAL | Before TPD | 110.500 | 11.085 | 0.45% | 0.611 |
| | After TPD | 111.000 | 10.806 | | |
| BMD | Before TPD | 0.736 | 0.172 | 20.04% | <0.001* |
| | After TPD | 0.884 | 0.194 | | |
| TBS˚ | Before TPD | 1.076 | 0.118 | 11.87% | 0.019* |
| | After TPD | 1.204 | 0.182 | | |

(*Continued*)

**Table 2.** (Continued)

| | | Mean/median | SD/IQR | Variation % | p-value |
|---|---|---|---|---|---|
| BSI° | Before TPD | 2.467 | 0.748 | -25.46% | 0.001* |
| | After TPD | 1.839 | 0.442 | | |

° = non-parametric distribution, with values presented as median with interquartile range (IQR) and compared with Wilcoxon signed rank test

* = statistically significant difference (p<0.05).

interactions of variables in living systems of the real world is completely lost. Auto-Cm system, a fourth generation ANN, arises just to overcome these limitations. Auto-CM has been applied in different medical contexts with interesting results demonstrating the ANNs' usefulness in easily "untangle the ball of yarn" of complex systems characterized by a lot of variables with different significance [34–38]. In our population, we separately analyzed the maps obtained before and after the treatment of TPD. The analysis clearly highlights a complex relationship between bone quantity and bone quality parameters, with high adaptive weight among the connections. When comparing pre- and post-treatment data in Tab. 1, we observe low absolute variations of bone geometry parameters' values, between 1 and 2%, but a noteable modification of the connection maps in ANNs. In particular, in the pre-therapy map the variables are divided into three leaves connected by two central hubs (Fig 1). They are HAL, indicating the length of femoral neck, proportionally related to fracture risk, and FS_BMD. In the post-therapy map (Fig 2) there is a change of the connections: FS_BMD migrates from central hub to periphery and leaves its position to NN_SECT_MOD, which is an index of resistance to compressive and flexural loads. This finding confirms the data of Stewart et al. and Jiang et al. [51,52], which demonstrated a positive effect of TPD on bone strength with an increase of CSA, that indicates a characteristic similar to SECT_MOD. Our data join to those described in the few papers published regarding this item in humans [30,32], confirming the known effect of TPD on the geometrical and structural bone parameters observed in animals [51,52]. Thus, ANNs maps' interconnections after TPD therapy change, with the grouping around the hub CSMI, expression of increase in resistance to compressive loads, while in pre-treatment the variables are more spread out.

Despite BSI did not modify its relationship with the other variables, it remains the index that shows the greatest percentage of variation (about 14%), suggesting a significant amelioration of bone strength. This might be due to the presence of lots of variables related to femur and just a few related to lumbar spine, that could explain an easy grow of the network affecting the same region and a different location on the map of lumbar variables.

Considering our four models, namely PreR, PostR, PreNR and PostNR,in the networks concerning the responders (Fig 3A and 3B) there is a high number of connections in the MRG: PreR shows 9 hubs and 22 connections, whereas PostR 10 hubs and 33 connections. This increase in the number of connections indicates an increase in the complexity of the system. In fact, there is an improvement of 50% of connections and 11% of related hubs, and an inclusion of the parameters referring to bone geometry not included in the PreR map (NN_CSMI and NN_SECT_MOD, Fig 3B). In constructions' science this is considered an increase of the resistance of the system (building resistance to collapse) [53].

A significant difference also exists comparing PreNR and PostNR (Fig 4A and 4B), because connections in the map increase significantly (400%), from 2 to 10, and there is a 133% increase in hubs, namely from 3 to 7. This indicates a marked gain in complexity after the treatment with TPD, including the cortical resistance parameters at all the considered femoral regions (Fig 4B).

**Table 3. SDI and DXA parameters of the "non-responders" (26 patients; 21 females and 5 males) before and after TPD therapy.**

| | | Mean/median | SD/IQR | Variation % | p-value |
|---|---|---|---|---|---|
| NN_BMD | Before TPD | 0.758 | 0.114 | 0.67% | 0.661 |
| | After TPD | 0.763 | 0.116 | | |
| NN_CSA | Before TPD | 2.345 | 0.428 | -0.38% | 0.775 |
| | After TPD | 2.336 | 0.417 | | |
| NN_CSMI˚ | Before TPD | 2.100 | 0.868 | -3.21% | 0.182 |
| | After TPD | 2.032 | 0.856 | | |
| NN_WIDTH | Before TPD | 3.255 | 0.309 | -0.89% | 0.327 |
| | After TPD | 3.226 | 0.321 | | |
| NN_SECT_MOD | Before TPD | 1.197 | 0.325 | -0.95% | 0.549 |
| | After TPD | 1.185 | 0.320 | | |
| NN_BR˚ | Before TPD | 12.763 | 3.969 | -3.77% | 0.131 |
| | After TPD | 12.281 | 3.523 | | |
| IT_BMD | Before TPD | 0.769 | 0.150 | -0.59% | 0.721 |
| | After TPD | 0.765 | 0.162 | | |
| IT_CSA | Before TPD | 4.133 | 0.993 | -0.94% | 0.567 |
| | After TPD | 4.094 | 1.009 | | |
| IT_CSMI˚ | Before TPD | 12.475 | 5.363 | -4.43% | 0.124 |
| | After TPD | 11.923 | 5.461 | | |
| IT_WIDTH | Before TPD | 5.629 | 0.577 | -0.10% | 0.933 |
| | After TPD | 5.624 | 0.591 | | |
| IT_SECT_MOD˚ | Before TPD | 3.490 | 1.256 | -4.99% | 0.066 |
| | After TPD | 3.316 | 1.277 | | |
| IT_BR | Before TPD | 10.553 | 2.421 | 2.04% | 0.310 |
| | After TPD | 10.768 | 2.794 | | |
| FS_BMD | Before TPD | 1.256 | 0.216 | -3.74% | 0.002* |
| | After TPD | 1.209 | 0.199 | | |
| FS_CSA | Before TPD | 3.595 | 0.713 | -3.49% | <0.01* |
| | After TPD | 3.470 | 0.681 | | |
| FS_CSMI˚ | Before TPD | 3.374 | 1.289 | -9.02% | 0.03* |
| | After TPD | 3.070 | 1.264 | | |
| FS_WIDTH | Before TPD | 3.004 | 0.238 | 0.26% | 0.608 |
| | After TPD | 3.012 | 0.266 | | |
| FS_SECT_MOD˚ | Before TPD | 2.171 | 0.715 | -9.94% | <0.01* |
| | After TPD | 1.955 | 0.742 | | |
| FS_BR˚ | Before TPD | 3.529 | 1.521 | -0.36% | 0.007* |
| | After TPD | 3.516 | 1.452 | | |
| SHAFT_NECK_ANGLE | Before TPD | 127.750 | 4.833 | 0.52% | 0.358 |
| | After TPD | 128.415 | 5.436 | | |
| SDI˚ | Before TPD | 7.500 | 6.500 | 13.33% | 0.109 |
| | After TPD | 8.500 | 6.250 | | |
| HAL | Before TPD | 106.269 | 10.034 | 0.25% | 0.560 |
| | After TPD | 106.538 | 10.033 | | |
| BMD | Before TPD | 0.750 | 0.128 | 2.19% | 0.077 |
| | After TPD | 0.767 | 0.134 | | |
| TBS˚ | Before TPD | 1.156 | 0.161 | -0.04% | 0.404 |
| | After TPD | 1.156 | 0.158 | | |

(*Continued*)

**Table 3.** (Continued)

| | | Mean/median | SD/IQR | Variation % | p-value |
|---|---|---|---|---|---|
| **BSI** | Before TPD | 2.413 | 0.668 | -6.57% | <0.01* |
| | After TPD | 2.255 | 0.657 | | |

° = non-parametric distribution, with values presented as median with interquartile range (IQR) and compared with Wilcoxon signed rank test

* = statistically significant difference (p<0.05)

Comparing the PreNR group's results with those of the PreR group, we noted 91% less connections and 67% less hubs interconnected in the PreNR group (2 connections and 3 hubs in the PreNR and 22 connections and 9 hubs in the PreR). Comparing PostNR with PostR there are 70% less connections and 30% less interconnected nodes. So, "non-responders" have less interconnections than "responders", both before and after drug therapy. An ANNs map with few connections seems to reflect a lower effect of TPD therapy, as indicated in literature [21].

This study points out the non-secondary role of DXA derived bone geometry parameters that are worth of a specific insight for their importance in identifying patients who are responsive or not to therapy. Another interesting finding is the reduction of BSI after TPD therapy, that suggests an increasing of bone strength. Thus, BSI appears to be a sensitive index of TPD effect, because it ameliorates even in the patients that do not present the expected relevant increase of BMD or of other DXA quality parameters. Finally, this study highlights the utility of the ANNs in the study of an item constituted by plenty of variables of different biological significance.

Limitation of this work is the not great number of cases studied, that suggests the need to extend this type of analysis to a larger group of patients. Another limitation of the study may be the lack of familiarity in the use of this new method of analysis which, however, is the basis of artificial intelligence which will increasingly tend to condition scientific activities as well.

Two conclusions arise from this study: In primis, TPD treatment appears to ameliorate not only bone quantity (BMD), but also bone texture and bone strain. Bone quality parameters (TBS, BSI, HSA), easily achieved by standard DXA scans, appear to be relevant in predicting the pharmacological response and are worthwhile of a greater consideration in clinical

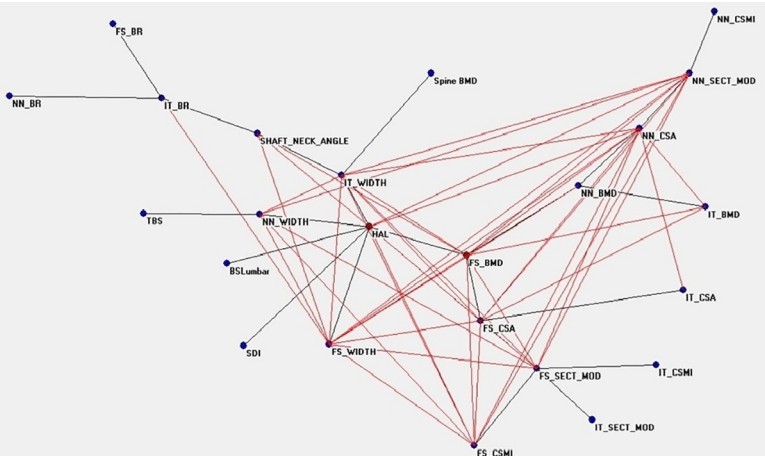

**Fig 1. Semantic map showing the relations between the investigated anagraphic, anthropometric, densitometric, biochemical and clinical parameters in the whole population before treatment.**

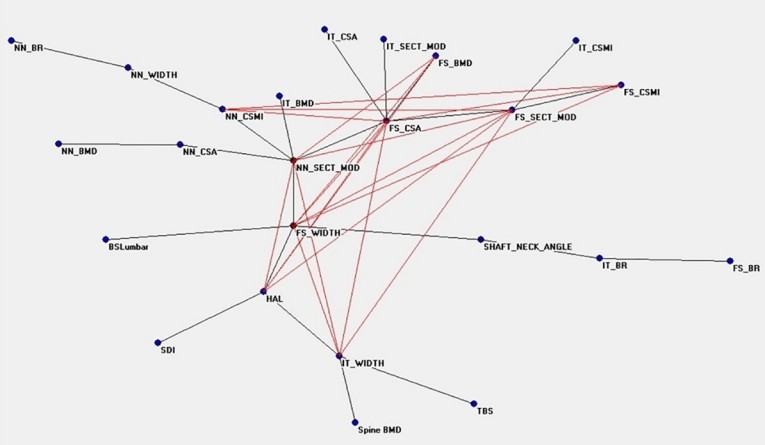

**Fig 2. Semantic map showing the relations between the investigated anagraphic, anthropometric, densitometric, biochemical and clinical parameters in the whole population after treatment.**

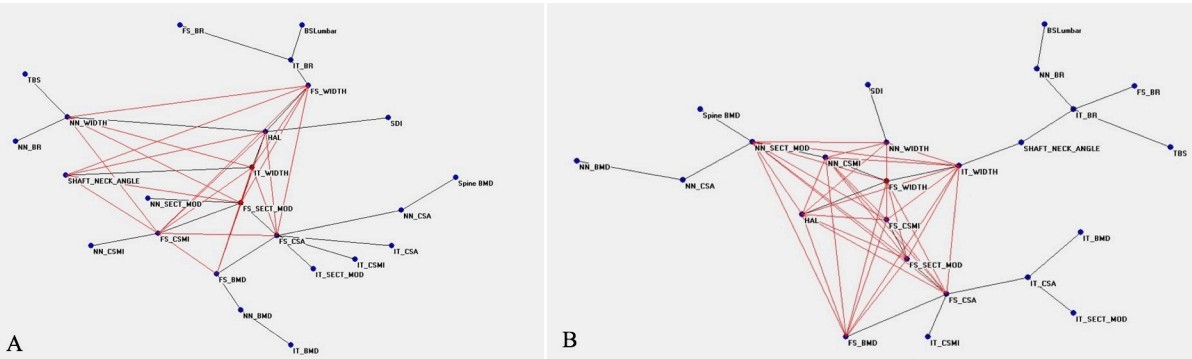

**Fig 3.** Semantic map showing the relations between the investigated anagraphic, anthropometric, densitometric, biochemical and clinical parameters in the "responders" before treatment (a) and after treatment (b).

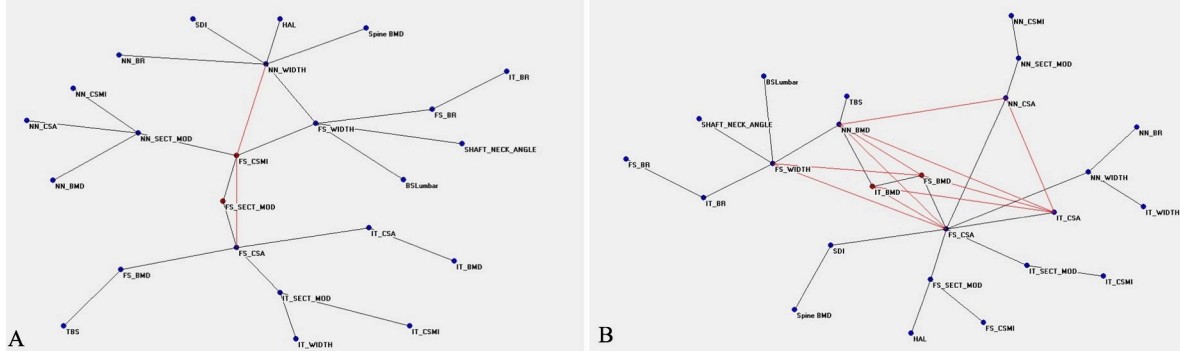

**Fig 4.** Semantic map showing the relations between the investigated anagraphic, anthropometric, densitometric, biochemical and clinical parameters in the "non-responders" before treatment (a) and after treatment (b).

practice. Secondly, ANNs proves itself to be useful in understanding the relations between variables of complex systems as those of multifactorial chronic diseases.

## Author Contributions

**Conceptualization:** Luca Petruccio Piodi, Luca Rinaudo, Fabio Massimo Ulivieri.

**Data curation:** Carmelo Messina, Luca Petruccio Piodi, Enzo Grossi, Cristina Eller-Vainicher, Maria Luisa Bianchi, Sergio Ortolani, Marco Di Stefano, Fabio Massimo Ulivieri.

**Formal analysis:** Carmelo Messina, Enzo Grossi, Luca Rinaudo, Fabio Massimo Ulivieri.

**Investigation:** Fabio Massimo Ulivieri.

**Methodology:** Luca Rinaudo, Fabio Massimo Ulivieri.

**Supervision:** Fabio Massimo Ulivieri.

**Validation:** Fabio Massimo Ulivieri.

**Visualization:** Fabio Massimo Ulivieri.

**Writing – original draft:** Luca Petruccio Piodi, Luca Rinaudo, Fabio Massimo Ulivieri.

**Writing – review & editing:** Carmelo Messina, Luca Petruccio Piodi, Luca Maria Sconfienza, Fabio Massimo Ulivieri.

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
