## [Decision Letter · Decision Letter 0]

31 Oct 2019

PONE-D-19-25972

BONE QUALITY DXA PARAMETERS IN FRACTURED OSTEOPOROTIC PATIENTS TREATED WITH TERIPARATIDE: STANDARD STATISTICAL AND ARTIFICIAL NEURAL NETWORK ANALYSIS

PLOS ONE

Dear Dr Ulivieri,

Thank you for submitting your manuscript to PLOS ONE. After careful consideration, we feel that it has merit but does not fully meet PLOS ONE’s publication criteria as it currently stands. Therefore, we invite you to submit a revised version of the manuscript that addresses the points raised during the review process.

Authors presented an interesting work. Although there are certain aspects that should be revised: material and methods description, include the study limitations and future work, etc. The major problem of the manuscript is that its purpose is not clear. The title is not in agreement with the aim and the conclusion. The results part linked to use of ANN is almost empty. Authors have to select the purpose of this manuscript and rewrite it accordingly.

We would appreciate receiving your revised manuscript by Dec 15 2019 11:59PM. To enhance the reproducibility of your results, we recommend that if applicable you deposit your laboratory protocols in protocols.io, where a protocol can be assigned its own identifier (DOI) such that it can be cited independently in the future. For instructions see: http://journals.plos.org/plosone/s/submission-guidelines#loc-laboratory-protocols

We look forward to receiving your revised manuscript.

Kind regards,

María Angeles Pérez, PhD

Academic Editor

PLOS ONE

Journal Requirements:

2. Please make sure you have thoroughly discussed any limitations of your study within the Discussion section. Additionally, please refer to any post-hoc corrections made during your statistical analysis for multiple comparisons. Please justify the reasons if these were not performed.

The authors have declared that no competing interests exist.

The engineer Luca Rinaudo, former working in Politecnico of Turin and now employed by the commercial company: "TECHNOLOGIC S.r.l”, has extracted and tabulated the densitometric data and has applied the mathematical algorithms based on the finite element analysis to calculate the Bone Strain Index. TECHNOLOGIC S.r.l. provided support in the form of salary for author LR, but did not have any role in the study design, data collection and analysis, decision to publish, or preparation of the manuscript.

Reviewers' comments:

Reviewer's Responses to Questions

**Comments to the Author**

1. Is the manuscript technically sound, and do the data support the conclusions?

Reviewer #1: Partly

Reviewer #2: Yes

2. Has the statistical analysis been performed appropriately and rigorously? 

Reviewer #1: Yes

Reviewer #2: Yes

3. Have the authors made all data underlying the findings in their manuscript fully available?

Reviewer #1: Yes

Reviewer #2: No

4. Is the manuscript presented in an intelligible fashion and written in standard English?

Reviewer #1: Yes

Reviewer #2: Yes

5. Review Comments to the Author

Reviewer #1: The aim of this study was to investigate the action of teriparatide on a new parameter of bone quality, namely Bone Strain Index (BSI), derived from DXA lumbar scan and based on a mathematical model called finite element method.

The manuscript is not easy to read because the purpose of this study is unclear but it is of interest for the community. Some issues have to be solved before considering it for publication.

MAJ 1: The title is not in agreement with the objective and the conclusion of the study. What is the purpose of the study: Evaluate the effect of TPTD on bone or the use of ANN to manage subjects under treatment? Authors should change the title or the objective of the study accordingly. The manuscript has to be rewritten depending on these changes.

Independetly of this point:

MAJ 2: In the introduction and the methods section, authors should explain for clinicians that are not specialist of ANN, what it is, how it works in a more detailed fashion.

MAJ 3: Regarding the use of ANN, in such methodology ANN have to be trained. How did you train your network? What cohort or dataset did you use to train it? Authors also need to give some information linked to the ANN structure (number of layers, number of neurons per layer, activation functions,…).

MAJ 4: In the results part, it is unclear for me what the purpose of the study is: Effect of TPTD or the use of ANN to assess TPTD treatment. If, the answer is ANN use, authors have to focus on the mapping and the associations obtained using ANN methodology. In any case, authors have to better explain results obtained using the ANN methodology which is almost inexistent. Authors should also display the strength of the associations between the studied parameters in the ANN mapping results. Did these associations (correlation between parameters) are in agreement with those obtained using the standard statistical approach?

Reviewer #2: Review Manuscript PONE-D-19-25972

The authors present an evaluation of the effect of Teriparatide as treatment for osteoporosis by the study of DXA parameters using standard statistic and artificial neuronal network analysis.

This is an interesting work that uses advance statistical analysis of different parameters obtained from DXA scans to evaluate the effect of a treatment for osteoporosis. This work provides information that might be useful for clinicians to address osteoporotic fractures prediction. However, manuscript summited is not ready to be published since there are some aspects to be considered. All details mentioned are referred to the pdf version:

-Page 13, Patients and Methods. It should be Materials and Methods

-Page 14, Methods. The authors could consider to change the title of this subsection to better describe the paragraph. For example, “DXA data acquisition”.

-Bone strain index (BSI). As the reviewer could see in the reference the BSI is a software that calculate strain and stress using finite element models. Nevertheless, it is not clear what is the BSI, what mechanical parameters include and what are the assumption in the models use for its calculation, i.e. linear elastic models? More information should be given of this parameter since it is not a regular parameter found in all DXA scans.

-Tables. For all tables the number of subjects (n) used in the analysis should be given. This is a useful information when statistic data is presented.

-Regarding the statistical analysis, in results for responders and non-responders groups are presented no information about how many women and men are in each group. Some parameters values might have significant differences between men and women. A comment about this aspect is more than welcome in the discussion of the paper.

-Page 18, “Furthermore, we divided… …”non-responders””. This information was already mentioned in the materials and methods section.

-Bone tissue, the author do not consider the different tissues, i.e. cortical and trabecular, in the statistical analysis. Trabecular and cortical bone are different and might have a difference response for the treatment that can be evaluated with the mechanical response. A comment addressing this aspect should be given at the discussion part.

-Discussion. In this section is missing that the authors highlight the importance of their findings.

-Page 24, “TPD is an… …the bone structure”. This sentence should be at the introduction not at the discussion part.

-Page 24, “The most important… … routine diagnostic practice.” Which is the added value of this sentence in this paragraph of the discussion?

-Page 24, “In fact, if bone strength ameliorates, BSI has to decrease.” Why does it has to decrease? A comment about that should be given.

-Page 24, “…of bone resistance to mechanical stresses.” A reference is missing

-Page 24-25, “It has de ability… …classical statistical approach.” This sentence should be at the introduction or at materials and methods.

-Page 25, “Despite BSI… … amelioration of bone strength.” Why?

Page 25, “When looking… ...,”post non-responders (PostNR)”.” This information should be at materials and methods.

Page 25-26, “In the networks… …33 connections.” This information fits better at the results sections than at the discussion section.

-At some point of the discussion the limitations for the analysis should be mentioned.

-Figures. It is difficult to differentiate the colour of some nodes and lines, when a zoom is applied the images get blurry.

-Figure legends, in general the caption of the figures should explain everything that the figure shows. In this sense, when a reader sees the figure and reads the caption the reader should not go to the text to understand what is in the figure.

The manuscript is well written and the study is really interesting for bone fracture community. The reviewer encourages the authors to consider all the changes mentioned previously. With the changes suggested in this review, the manuscript will be ready for publication.

6. PLOS authors have the option to publish the peer review history of their article (what does this mean?). If published, this will include your full peer review and any attached files.

Reviewer #1: No

Reviewer #2: No

---

## [Author Response · Author response to Decision Letter 0]

23 Jan 2020

We thank the reviewers for their criticisms regarding our manuscript PONE-D-19-25972

“BONE QUALITY DXA PARAMETERS IN FRACTURED OSTEOPOROTIC PATIENTS TREATED WITH TERIPARATIDE: STANDARD STATISTICAL AND ARTIFICIAL NEURAL NETWORK ANALYSIS”

Reviewer #1: The aim of this study was to investigate the action of teriparatide on a new parameter of bone quality, namely Bone Strain Index (BSI), derived from DXA lumbar scan and based on a mathematical model called finite element method.

The manuscript is not easy to read because the purpose of this study is unclear but it is of interest for the community. Some issues have to be solved before considering it for publication.

MAJ 1: The title is not in agreement with the objective and the conclusion of the study. What is the purpose of the study: Evaluate the effect of TPTD on bone or the use of ANN to manage subjects under treatment? Authors should change the title or the objective of the study accordingly. The manuscript has to be rewritten depending on these changes.

RESPONSE: We have focused the dissertation on the use of ANNs to manage the TPD effect on bone quantity and quality in osteoporosis. Consequently we have modified the title.

Independently of this point:

MAJ 2: In the introduction and the methods section, authors should explain for clinicians that are not specialist of ANN, what it is, how it works in a more detailed fashion.

RESPONSE: We have expanded in the introduction and methods section the explanation about Auto-CM system, the ANN employed in our paper.

MAJ 3: Regarding the use of ANN, in such methodology ANN have to be trained. How did you train your network? What cohort or dataset did you use to train it? Authors also need to give some information linked to the ANN structure (number of layers, number of neurons per layer, activation functions,…).

RESPONSE: Auto-CM is a special kind of unsupervised neural network which requires a training phase necessary to learn how variables are interconnected, as explained in the expanded and revised version of methods with a specific protocol: The learning algorithm of CM may be summarized in four orderly steps: a) signal transfer from the input into the hidden layer; b) adaptation of the connections value between the Input layer and the hidden layer; c) signal transfer from the hidden layer into the output layer; d) adaptation of the connections value between the hidden layer and the output layer. Since the learning is unsupervised the ANN uses all data available in our data set, not requiring a splitting for training testing protocol, generally used for supervised ANNs.

MAJ 4: In the results part, it is unclear for me what the purpose of the study is: Effect of TPTD or the use of ANN to assess TPTD treatment. If, the answer is ANN use, authors have to focus on the mapping and the associations obtained using ANN methodology. In any case, authors have to better explain results obtained using the ANN methodology which is almost inexistent. Authors should also display the strength of the associations between the studied parameters in the ANN mapping results. Did these associations (correlation between parameters) are in agreement with those obtained using the standard statistical approach? 

RESPONSE: We have revised the results and discussion sections in order to better focusing ANNs implication in this item. We have found associations between the studied parameters that are in agreement both in classical statistical analysis and in ANNs . We attach the excel format with these data in the new submission as a supplemental file.

Reviewer #2: Review Manuscript PONE-D-19-25972

The authors present an evaluation of the effect of Teriparatide as treatment for osteoporosis by the study of DXA parameters using standard statistic and artificial neuronal network analysis.

This is an interesting work that uses advance statistical analysis of different parameters obtained from DXA scans to evaluate the effect of a treatment for osteoporosis. This work provides information that might be useful for clinicians to address osteoporotic fractures prediction. However, manuscript summited is not ready to be published since there are some aspects to be considered. All details mentioned are referred to the pdf version:

-Page 13, Patients and Methods. It should be Materials and Methods

RESPONSE: We should prefer to maintain the term “Patients” being our work a study on humans

-Page 14, Methods. The authors could consider to change the title of this subsection to better describe the paragraph. For example, “DXA data acquisition”.

RESPONSE: We have followed the reviewer’s suggestion.

-Bone strain index (BSI). As the reviewer could see in the reference the BSI is a software that calculate strain and stress using finite element models. Nevertheless, it is not clear what is the BSI, what mechanical parameters include and what are the assumption in the models use for its calculation, i.e. linear elastic models? More information should be given of this parameter since it is not a regular parameter found in all DXA scans.

RESPONSE: We have added a paragraph ad hoc in the manuscript.

Tables. For all tables the number of subjects (n) used in the analysis should be given. This is a useful information when statistic data is presented.

RESPONSE: We have added what requested.

-Regarding the statistical analysis, in results for responders and non-responders groups are presented no information about how many women and men are in each group. Some parameters values might have significant differences between men and women. A comment about this aspect is more than welcome in the discussion of the paper.

RESPONSE: We thank the reviewer for this specific comment, as effectively we noticed that the percentage of gender composition was different between the two groups. Non responder group was mainly composed by women (about 80%), while responder group was quite balanced. We highlighted this in the result section and further commented it in the discussion, as this may be one possible and additional reason for the change in BSI that deserves further investigation.

-Page 18, “Furthermore, we divided… …”non-responders””. This information was already mentioned in the materials and methods section.

RESPONSE: We have deleted the repetition.

-Bone tissue, the author do not consider the different tissues, i.e. cortical and trabecular, in the statistical analysis. Trabecular and cortical bone are different and might have a difference response for the treatment that can be evaluated with the mechanical response. A comment addressing this aspect should be given at the discussion part.

RESPONSE: Sorry, we have performed DXA scans at lumbar spine, composed by about 75% of trabecular bone, and femur, composed by about 50% of cortical bone. TBS and BSI are derived only from spine scan and not femur scan and hip structural analysis is applied only to femur and not to spine.

-Discussion. In this section is missing that the authors highlight the importance of their findings.

RESPONSE: We added a notation in this sense in the text.

-Page 24, “TPD is an… …the bone structure”. This sentence should be at the introduction not at the discussion part.

RESPONSE: We have modified the sentence.

-Page 24, “The most important… … routine diagnostic practice.” Which is the added value of this sentence in this paragraph of the discussion?

RESPONSE: We have deleted the indicated sentences.

-Page 24, “In fact, if bone strength ameliorates, BSI has to decrease.” Why does it has to decrease? A comment about that should be given.

RESPONSE: We have added a comment ad hoc in the text.

-Page 24, “…of bone resistance to mechanical stresses.” A reference is missing

RESPONSE: We have added the references requested

-Page 24-25, “It has de ability… …classical statistical approach.” This sentence should be at the introduction or at materials and methods.

RESPONSE: We have deleted the sentence. The concept is already exhaustively explained in the Introduction and Method sections.

-Page 25, “Despite BSI… … amelioration of bone strength.” Why?

RESPONSE: We have added in the text a possible explanation

Page 25, “When looking… ...,”post non-responders (PostNR)”.” This information should be at materials and methods.

RESPONSE: We have followed the suggestions modifying the text. 

Page 25-26, “In the networks… …33 connections.” This information fits better at the results sections than at the discussion section.

RESPONSE: We prefer to leave this evidence here, in order to avoid the overloading of not yet explained data in the Results section.

-At some point of discussion the limitations for the analysis should be mentioned.

RESPONSE: We have added the requested limitation of the study in the text

-Figures. It is difficult to differentiate the colour of some nodes and lines, when a zoom is applied the images get blurry. 

RESPONSE: We have tried to make the figures as readable as possible

-Figure legends, in general the caption of the figures should explain everything that the figure shows. In this sense, when a reader sees the figure and reads the caption the reader should not go to the text to understand what is in the figure.

RESPONSE: We share the observation, but each figure contains more than twenty variables and their list, already reported in Method section, would occupy much space of the journal pages for each figure.

The manuscript is well written and the study is really interesting for bone fracture community. The reviewer encourages the authors to consider all the changes mentioned previously. With the changes suggested in this review, the manuscript will be ready for publication.

---

## [Decision Letter · Decision Letter 1]

27 Jan 2020

PONE-D-19-25972R1

ARTIFICIAL NEURAL NETWORK ANALYSIS OF BONE QUALITY DXA PARAMETERS RESPONSE TO TERIPARATIDE IN FRACTURED OSTEOPOROTIC PATIENTS

PLOS ONE

Dear Dr Ulivieri,

Thank you for submitting your manuscript to PLOS ONE. After careful consideration, we feel that it has merit but does not fully meet PLOS ONE’s publication criteria as it currently stands. Therefore, we invite you to submit a revised version of the manuscript that addresses the points raised during the review process.

Authors should additionally improve the discussion and extend the limitations of the work. 

We would appreciate receiving your revised manuscript by Mar 06 2020 11:59PM. To enhance the reproducibility of your results, we recommend that if applicable you deposit your laboratory protocols in protocols.io, where a protocol can be assigned its own identifier (DOI) such that it can be cited independently in the future. For instructions see: http://journals.plos.org/plosone/s/submission-guidelines#loc-laboratory-protocols

We look forward to receiving your revised manuscript.

Kind regards,

María Angeles Pérez, PhD

Academic Editor

PLOS ONE

Reviewers' comments:

Reviewer's Responses to Questions

**Comments to the Author**

1. If the authors have adequately addressed your comments raised in a previous round of review and you feel that this manuscript is now acceptable for publication, you may indicate that here to bypass the “Comments to the Author” section, enter your conflict of interest statement in the “Confidential to Editor” section, and submit your "Accept" recommendation.

Reviewer #1: All comments have been addressed

Reviewer #2: All comments have been addressed

2. Is the manuscript technically sound, and do the data support the conclusions?

Reviewer #1: Partly

Reviewer #2: Yes

3. Has the statistical analysis been performed appropriately and rigorously? 

Reviewer #1: Yes

Reviewer #2: Yes

4. Have the authors made all data underlying the findings in their manuscript fully available?

Reviewer #1: Yes

Reviewer #2: Yes

5. Is the manuscript presented in an intelligible fashion and written in standard English?

Reviewer #1: Yes

Reviewer #2: Yes

6. Review Comments to the Author

Reviewer #1: Dear Authors,

The quality of the revized manuscript has greatly increased.

However, I still have some concerns:

First: The title and the discussion orientation are not completly inline. Auhtors should focus their discussion on the use of ANN as an alternative of standard statistics in order to agree with the title of the manuscript. Instead, the discussion is focussed on the evalution of BMD, TBS and HSA parameters.

Second: The limitation part is really brief and need to be developped.I supposed that technical limitations linked to the use of ANN exist and hve to be mentioned. Please, develop this part.

Third: The conclusion of the manuscript has to be focused on the use of ANN and not to the variations of BMD, TBS,BSI or HSA parameters to be in agreement with the title of the manuscript. Please, update the conclusion.

Reviewer #2: Review Manuscript PONE-D-19-25972R1

The authors fairly answered all the quotes presented in the previous revision and clarified all the doubts. The manuscript was changed according to the suggestions made. The new manuscript is quite better than it previous version. From the reviewer point of view, the paper is ready for publication.

7. PLOS authors have the option to publish the peer review history of their article (what does this mean?). If published, this will include your full peer review and any attached files.

Reviewer #1: No

Reviewer #2: No

---

## [Author Response · Author response to Decision Letter 1]

12 Feb 2020

We thank the reviewers for their criticisms regarding our manuscript PONE-D-19-25972R1

“ARTIFICIAL NEURAL NETWORK ANALYSIS OF BONE QUALITY DXA PARAMETERS RESPONSE TO TERIPARATIDE IN FRACTURED OSTEOPOROTIC PATIENTS”

Reviewer #1:

Dear Authors,

The quality of the revized manuscript has greatly increased. However, I still have some concerns:

First: The title and the discussion orientation are not completly inline. Auhtors should focus their discussion on the use of ANN as an alternative of standard statistics in order to agree with the title of the manuscript. Instead, the discussion is focussed on the evalution of BMD, TBS and HSA parameters.

RESPONSE: We have implemented the discussion as requested.

Second: The limitation part is really brief and need to be developped.I supposed that technical limitations linked to the use of ANN exist and hve to be mentioned. Please, develop this part.

RESPONSE: We have developed the limitation paragraph.

Third: The conclusion of the manuscript has to be focused on the use of ANN and not to the variations of BMD, TBS,BSI or HSA parameters to be in agreement with the title of the manuscript. Please, update the conclusion.

RESPONSE: We have better focused the conclusion paragraph.

Reviewer #2: Review Manuscript PONE-D-19-25972R1

The authors fairly answered all the quotes presented in the previous revision and clarified all the doubts. The manuscript was changed according to the suggestions made. The new manuscript is quite better than it previous version. From the reviewer point of view, the paper is ready for publication.

RESPONSE: We thank the Reviewer for his appreciation.

---

## [Editor Report · Decision Letter 2]

18 Feb 2020

ARTIFICIAL NEURAL NETWORK ANALYSIS OF BONE QUALITY DXA PARAMETERS RESPONSE TO TERIPARATIDE IN FRACTURED OSTEOPOROTIC PATIENTS

PONE-D-19-25972R2

Dear Dr. Ulivieri,

We are pleased to inform you that your manuscript has been judged scientifically suitable for publication and will be formally accepted for publication once it complies with all outstanding technical requirements.

With kind regards,

María Angeles Pérez, PhD

Academic Editor

PLOS ONE
---

## [Editor Report · Acceptance letter]

2 Mar 2020

PONE-D-19-25972R2 

Artificial neural network analysis of bone quality DXA parameters response to teriparatide in fractured osteoporotic patients 

Dear Dr. Ulivieri:

I am pleased to inform you that your manuscript has been deemed suitable for publication in PLOS ONE. Congratulations! Your manuscript is now with our production department. 

With kind regards,

on behalf of

Dr. María Angeles Pérez 

Academic Editor

PLOS ONE